# Neuroprotective Effect of Terpenoids Recovered from Olive Oil By-Products

**DOI:** 10.3390/foods10071507

**Published:** 2021-06-29

**Authors:** Zully J. Suárez Montenegro, Gerardo Álvarez-Rivera, José David Sánchez-Martínez, Rocío Gallego, Alberto Valdés, Mónica Bueno, Alejandro Cifuentes, Elena Ibáñez

**Affiliations:** Foodomics Laboratory, Institute of Food Science Research (CIAL, CSIC), Nicolas Cabrera, 9, 28049 Madrid, Spain; zully.suarez.montenegro@cial.uam-csic.es (Z.J.S.M.); gerardo.alvarez@csic.es (G.Á.-R.); jd.sanchez.martinez@csic.es (J.D.S.-M.); rocio.gallego@csic.es (R.G.); a.valdes@csic.es (A.V.); monibuenofdez@gmail.com (M.B.); elena.ibanez@csic.es (E.I.)

**Keywords:** olive oil by-products, terpenes fractionation, adsorbent-assisted processes, supercritical CO_2_ extraction, Alzheimer’s disease, neuroprotective effect

## Abstract

The neuroprotective potential of 32 natural extracts obtained from olive oil by-products was investigated. The online coupling of supercritical fluid extraction (SFE) and dynamic adsorption/desorption allowed the selective enrichment of olive leaves extracts in different terpenoids’ families. Seven commercial adsorbents based on silica gel, zeolite, aluminum oxide, and sea sand were used with SFE at three different extraction times to evaluate their selectivity towards different terpene families. Collected fractions were analyzed by gas chromatography coupled to quadrupole-time-of-flight mass spectrometry (GC-QTOF-MS) to quantify the recoveries of monoterpenes (C10), sesquiterpenes (C15), diterpenes (C20), and triterpenes (C30). A systematic analysis of the neuroprotective activity of the natural extracts was then carried out. Thus, a set of in vitro bioactivity assays including enzymatic (acetylcholinesterase (AChE), butyrylcholinesterase (BChE)), and anti-inflammatory (lipoxidase (LOX)), as well as antioxidant (ABTS), and reactive oxygen and nitrogen species (ROS and RNS, respectively) activity tests were applied to screen for the neuroprotective potential of these extracts. Statistical analysis showed that olive leaves adsorbates from SS exhibited the highest biological activity potential in terms of neuroprotective effect. Blood–brain barrier permeation and cytotoxicity in HK-2 cells and human THP-1 monocytes were studied for the selected olive leaves fraction corroborating its potential.

## 1. Introduction

The valorization of biological wastes from agricultural activity and processing industries has become a challenge for science, which seeks to investigate new alternatives for the sustainable management of the growing waste generated. Olive leaves (*Olea europaea* L.) are an important agricultural residue in Spain and other olive oil producing countries, representing 25 percent of the total biomass generated in the olive oil industry [1], and accounting for over 500,000 t per year in Spain [2]. Olive leaves waste refer to the residues coming from both the mechanical pruning of the trees and the cleaning processes of the olive during the harvest of the fruit.

An important strategy to revalorize this food by-product is to extract compounds with interesting health properties. Several studies have reported important bioactive properties of compounds from olive leaves [3], thus being considered as a promising natural source of biologically active compounds for the food, cosmetic, and pharmaceutical industries. The bioactivity of olive leaves has been traditionally associated to its content in phenolic derivatives [4,5,6] and flavonoids and terpenoids [7,8,9,10]. These compounds provide a wide range of health properties [11,12,13], such as anti-inflammatory [12,13], antioxidant [11,14,15,16], and antiproliferative activities against cancer cell lines [11,17].

Alzheimer’s disease (AD) is the most common cause of dementia (60 to 80 percent of cases) and death among older people [18,19], representing a huge economic and social cost for sanitary systems around the world due to an increase in the aging population [20,21].

The hallmark pathologies of AD are a deficit in the cholinergic transmission and the accumulation of both extracellular toxic plaques of amyloid-β (Aβ) protein and intracellular hyperphosphorylated Tau (τ) neurofibrillary tangles (NFTs) [22,23,24,25,26] due to genetic, environmental and nutritional factors, oxidative stress [19,27,28], and neuroinflammation [21,22,29], among other factors. Neuroinflammation response is closely associated with oxidative stress [27]. Reactive oxygen/nitrogen species (ROS/RNS) including superoxide, hydroxyl radicals, hydrogen peroxide, and nitric oxide (NO) [30,31,32] are normal byproducts of oxidation of brain lipids, carbohydrates, proteins and DNA [27], and they have been connected to the presence of the Aβ plaques and NFTs in AD [33]. Therefore, the most important challenge in the search for a novel (and successful) treatment against AD has been the multifactorial nature of this pandemic.

The only current drugs approved for therapeutic treatment of AD act as cholinergic enzyme inhibitors and come from alkaloid secondary metabolites, which are considered the most dangerous end of the phytochemical spectrum [34]. For this reason, the study of neuroprotective effects from other secondary metabolites that can be obtained from low-cost natural sources such as food by-products, and in our case olive leaves is an interesting topic of research. In this regard, phenolic derivatives from olive leaves have been deeply investigated [35,36,37,38,39,40], and in a less extent other compounds such as terpenoids [40,41,42]. These compounds have been shown to provide a wide range of health properties [14,43], such as anti-inflammatory [44,45], antioxidant [15,16,46], and anticancer [11,17,47,48].

In order to obtain interesting bioactive compounds from olive leaves waste with high-added value in a sustainable way, alternative extraction technologies are required to substitute the conventional extraction procedures. In this work, a methodology that combines supercritical fluid extraction (SFE) and adsorption, and recently described by Suárez-Montenegro et al. [49], was used. Moreover, a new adsorbent (sea sand, SS) was tested to obtain fractions from olive leaves enriched in terpenoids.

In summary, the aim of this work was to combine SFE with dynamic and low-cost adsorption/desorption processes in order to obtain different terpenes fractions from olive leaves wastes that could provide differential biological neuroprotective activities. This work focuses on the evaluation of the neuroprotective potential of terpenes from olive leaves extracts by means of a set of in vitro activity assays (that include enzymatic (acetylcholinesterase (AChE), butyrylcholinesterase (BChE)), anti-inflammatory (lipoxidase (LOX)), as well as antioxidant (ABTS), and ROS and RNS activity tests) together with blood–brain barrier (BBB) permeation experiments and citotoxicity in HK-2 cells and human THP-1 monocytes.

## 2. Materials and Methods

### 2.1. Vegetable Material

Shade-dried olive leaves (Cornicabra variety) with humidity lower than ten percent were supplied by a local producer (Murciana de Herboristería S.A., Murcia, Spain). Pre-treatment of samples consisted of manually removing branches and other impurities from leaves before grinding with a knife mill (Retsch Grindomix Ref GM200-Germany) at 8000 rpm for 40 s. Sieving to 500–1000 µm particle size was done using an electromagnetic sieve shaker (CISA Sieving Technologies BA-200N, Barcelona, Spain).

### 2.2. Adsorbent Material

Different types of adsorbents were tested. Pore size, particle size and surface area of studied adsorbents (Silica gel (S60), Silica gel (S60P), Silica gel (S150), Silica gel (S150P), Zeolite Y Ammonium (ZeAmG) and Aluminum oxide 150 Type T (AO)) are summarized in [49] as well as the SFE-adsorption/desorption conditions. Moreover, in this work, an additional adsorbent, SS with a particle size of 1000–6000 µm and a bulk density of 1818.5 mg/cm^3^, was included.

### 2.3. Chemical Reagents

Acetylcholinesterase (AChE) from *Electrophorus electricus* (electric eel) type VI-S, butyrylcholinesterase (BChE) from equine serum, and 2,2-azino-bis (3-ethylbenzothiazoline-6-sulphonic acid) (ABTS^•+^) were purchased from Sigma-Aldrich (Madrid, Spain). Trizma hydrochloride (Tris-HCl), bovine serum albumin (BSA), (KH_2_PO_4_) ≥ 99.0%, (NaH_2_PO_4_) ≥ 99.0%, (K_2_S_2_O_8_) ≥ 99.0%, (Na_2_CO_3_) ≥ 99.0%, sodium nitroprusside dehydrate (SNP), fluorescein sodium salt, sulphanilamide, naphthylethylene diamine dihydrochloride, phosphoric acid, gallic acid, quercetin, cholesterol, n-dodecane, linoleic acid (LA), porcine polar brain lipid (PBL), a PAMPA-BBB 96-well donor plate (MAIPNTR10), and 96-well acceptor plate (MATRNPS50) were acquired from Sigma-Aldrich (Madrid, Spain); 7-fluorobenzofurazan-4-sulfonamide (ABD-F) 98% was purchased from Alfa Aesar (Kandel, Germany); (±)-6-Hydroxy-2,5,7,8-tetramethylchromane-2-carboxylic acid (Trolox) > 97%, acetylthiocholine iodide (ATCI) ≥ 99.0%, butyrylthiocoline iodide (BTCI) ≥ 99.0%, and lipoxidase from glycine max (soybean), Type 1-B, were obtained from Sigma-Aldrich (Madrid, Spain); ethanol (EtOH) HPLC-grade was purchased from VWR Chemicals (Barcelona, Spain); galantamine hydrobromide, purity > 98.0%, and 2,2-azobis(2-amidinopropane) dihydrochloride (AAPH) were purchased from TCI Chemicals (Tokyo, Japan); and ultrapure water (18.2 MΩ cm) was obtained from a Millipore system (Billerica, MA, USA). All the 96-well microplate assays were performed in a spectrophotometer and fluorescent reader (Synergy HT, BioTek Instruments, Winooski, VT, USA). Griess reagent was prepared with 500 mg sulphanilamide, 50 mg naphthylethylene diamine dihydrochloride, and 1.25 mL of phosphoric acid in 48.5 mL of water.

### 2.4. Supercritical Fluid Extraction with Adsorption/Desorption

The extraction was carried out in a Speed Helix supercritical fluid extractor from Applied Separations (Allentown, PA, USA) using neat CO_2_ (Carburos Metálicos, Air Products Group, Madrid, Spain) as extraction solvent. The conditions for SFE-adsorption/desorption were based on a previous paper [49]. In addition to the adsorbents described in that paper and given in Section 2.2, in the present work we also tested SS as adsorbent. A brief description of the process is provided and a scheme of the system used is shown in Figure 1.

The adsorbent-assisted supercritical CO_2_ extraction process was carried out at 30 MPa and 60 °C dynamically for 120 min. The adsorbent material was placed in a stainless-steel cylindrical adsorption cell (29 cm length and 0.65 cm i.d., for a total column volume of 38.5 cm^3^) after the extraction cell, as shown in Figure 1; adsorbents were packed into the adsorption column with glass wool and high-quality cellulose disk filters were located at the entrance and exit of the column to prevent plugging. Carbon dioxide passed through the supercritical extraction cell and the extracted solute(s) was adsorbed dynamically by the packed material in the adsorption column; the whole process was carried out at the same *p* and T, that is, 30 Mpa and 60 °C. Fractions were collected every 20 min at the exit of the adsorption column, after depressurization through an expansion valve (Parker Autoclave Engineers, Erie, PA, USA). After 120 min, complete depressurization of the system was carried out for a total of 30 min.

Non-desorbed compounds remaining in the adsorbent after the whole process were recovered by washing the material with high-purity grade ethanol (VWR Chemicals-BDH, Barcelona, Spain) by agitation at room temperature for 2 h and filtered (they were called adsorbates). This treatment was applied for all materials except for zeolite that required a further centrifugation step at 10,000 rpm for 10 min. The supernatant was passed through a filter of 0.45 µm pore size and 13 mm diameter. All experiments were done in duplicate. Extraction yield (%) of all fractions was expressed on a dry weight basis. Recovery values (%) for a particular family of terpenes (Ci = C10, C15, C20, or C30) obtained at a defined extraction time (*t* = (0–20), (20–40), (40–60), (60–80), (80–100]) or (100–120) min), using a certain type of adsorbent material (*s* = SS, S60, S60P, S150, S150P, ZeAmG, or AO) was calculated according to Equation (1) as follows:(1)$$\%\;\mathrm{Recovery}\;\left[\;C_{i\left(t,s\right)}\right]=\frac{A_{ci\left(t,s\right)}}{\sum^{\;}A_{ci\left(control\right)}}\times 100$$
where *A_ci(t,s)_* is the abundance of the target terpenes family extracted under fixed conditions of time and adsorbent; and ∑*A_ci(control)_* is the sum of abundances of all terpene families (total terpenes abundance) obtained under control conditions (*t* = 120 min, without adsorbent).

From all samples collected during the kinetic process plus adsorbates, a total of 32 natural extracts from olive leaves enriched in terpenoids were obtained. This number comes from the 7 adsorbents and control (no adsorbent) at 3 points of kinetic extraction (*t* = (0–20), (40–60), and (100–120) min) tested, plus, the 7 adsorbates fractions that remained in the adsorbent and a global control (total extraction time = 120 min) obtained by SFE without using any adsorbent.

### 2.5. GC-QTOF-MS Analysis of Terpenoids

The analysis of the 32 extracts was performed employing an Agilent 7890B gas chromatography (GC) system coupled to an Agilent 7200 quadrupole time-of-flight (Q-TOF) mass spectrometer, equipped with an electronic ionization (EI) interface. The separation was carried out using an Agilent Zorbax DB5-MS Column (30 m × 250 μm i. D. × 0.25 μm) + 10 m DuraGuard capillary column. Helium was used as carrier gas at a constant flow rate of 0.8 mL/min. The injection volume was 1 μL. Splitless mode was used for injection, keeping the injector temperature at 250 °C. The GC oven was programmed at 60 = °C for 1 min, then increased at a rate of 10 °C/min to 325 °C, and held at this temperature for 10 min. MS detector was operated in full-scan acquisition mode at a m/z scan range of 50–600 Da (5 spectra per second). The temperatures of the transfer line, the quadrupole, and the ion source were set at 290, 150, and 250 °C, respectively.

Target terpenes were annotated by systematic mass spectra deconvolution and searched in the MS database, using the Agilent MassHunter Unknowns Analysis tool and NIST MS database search. A total of 40 terpenes and terpenoids were tentatively identified on the basis of the positive match of the experimental mass spectra with MS databases (i.e., NIST and Fiehn lib), exact mass values as determined by HRMS, data reported in literature, and commercial standards when available. GC-QTOF-MS parameters such as retention time, generated molecular formula, match factor values from the MS database search, and main HRMS fragments were considered for annotation. Identification reliability was considered satisfactory for chemical structures, showing math factor values above 70. Terpenoids such as thymol, squalene, phytol, alpha-tocopherol, alpha-amyrin, uvaol, and erythrodiol were confirmed with reference standard. More information about the structural elucidation of target terpenoids can be found in our recently reported paper by Suárez et al. (2021) [49].

### 2.6. In Vitro Bioactivity Assays

The natural extracts obtained from olive leaves were studied by a battery of in vitro assays related to AD, such as reduction of cholinergic (AChE), anti-inflammatory (LOX), and antioxidant (ABTS) activities. Inhibition of BChE, ROS, and RNS scavenging capacity, PAMPA-BBB permeability, and cytotoxicity studies with cells were performed using the extract that provided the best results for the AChE, LOX, and ABTS screening study.

#### 2.6.1. Anti-Cholinergic Activity Assay

Anti-cholinergic activity was evaluated by means of AChE and BChE inhibitory capacity enzymes of the extracts and based on Ellman’s method, modified by a fluorescent enzyme kinetics study using ABD-F as a fluorescent probe [50]. Previously, Michaelis–Menten constant (KM value) was measured to fix the substrate concentration at which the reaction rate is half of the maximum velocity rate. Concentrated stocks of both enzymes were prepared in buffer 150 mM Tris-HCl at pH 8.0 and 0.1% BSA was added to keep the stock solution stable. The microplate filling distribution for AChE/BChE inhibition assay was as follows: 100 μL of buffer (150 mM Tris-HCl at pH 8.0), 50 μL of ATCI (or BTCI) at a concentration of the KM value in ultrapure water, 100 μL of olive leaves extract at seven different concentrations (50 to 500 μg/mL for AChE and 37 to 370 for BChE) in EtOH/H_2_O (1:1, *v*/*v*). After 10 min of incubation, 25 μL of ABD-F (125 μM) in buffer was added and 25 μL of AChE (or BChE) was diluted at 0.8 U/mL in the buffer. Galantamine was used as a positive control standard and well without inhibitors as negative control. The fluorescence kinetic measurement was done in a microplate reader at λexcitation = 389 nm and λemission = 513 nm, runtime 15 min at intervals of 1 min and 37 °C set temperature, to reach Vmean of enzymatic reaction. Equation (2) represents the percentage of inhibition of the sample compared with the negative control, where V_1_ and V_0_ are mean velocity obtained for AChE (or BChE) in the presence and absence of galantamine or olive leaves extracts. Each measurement was carried out in triplicate and results are expressed as mean ± standard deviation. IC_50_ value represents concentration (µg/mL) of galantamine or olive leaves extract that produced 50 percent of cholinergic enzyme inhibition capacity compared with the control (without inhibitors); therefore, lower IC_50_ concentrations exhibited a major inhibitory potency than higher IC_50_ values.
(2)$$\%\;Inh=\frac{V_{0}-V_{1}\;}{V_{0}}\times 100$$

#### 2.6.2. Antioxidant and Scavenging Radical Capacity Assays: ABTS, ROS, and RNS

The antioxidant and scavenging activities were determined using ABTS^•+^, ORAC, and nitric oxide radical assays as described below.

ABTS^•+^ scavenging capacity

Antioxidant activity of all olive leaves extracts was measured using the ABTS^•+^ radical scavenging assay according to Re et al., (1999) [51] with modifications. Previously, the ABTS^•+^ radical was produced by the reaction of ABTS stock solution (7 mM) with 2.45 mM of potassium persulfate in the dark at room temperature during the 16 h before use. After this time, radical ABTS^•+^ solution was diluted with the buffer (5 mM, pH 7.4) to adjust the absorbance to 0.700 ± 0.002 measured at 734 nm wavelength and 30 °C. The 96-well microplate containing 250 μL ABTS^•+^ solution and 100 μL of olive leaves extract at six different concentrations (from 14.3 to 143 μg/mL EtOH/H_2_O (1:1, *v*/*v*)) was incubated in the dark at 30 °C. After 45 min, the absorbance at 734 nm was measured. Trolox and ascorbic acid were used as reference standards while rosemary extract was used as natural reference standard. All measurements were performed in triplicate. ABTS^•+^ inhibition percentage from olive leaves extracts was calculated as follows:(3)$$\%Inh=\frac{A_{ABTScontrol}-(A_{sample}-A_{sample\;blank})}{A_{ABTS\;control}}\times 100$$
where *A_ABTScontrol_* is the absorbance of ABTS^•+^ radical in buffer at *t* = 0 min; *A_sample_*, is the absorbance of an ABTS^•+^ solution mixed with extracts; and *A_sample blank_* is the absorbance of samples without ABTS^•+^. Comparisons between extracts were conducted by IC_50_ value calculated as the concentration (μg/mL) of OLE that inhibited 50% of the ABTS^•+^ radical.

Reactive Oxygen Species (ROS) scavenging capacity

Oxygen radical absorbance capacity (ORAC) method was performed as reported by Ou et al. (2001) [52]. Briefly, 100 μL of the extract sample at different concentrations (5–50 μg/mL) in EtOH/H_2_O (1: 9, *v*/*v*), 100 μL of AAPH (590 mM) in PBS buffer (30 mM, pH = 7.5), 25 μL of fluorescein (10 μM) in PBS buffer, and 100 μL of PBS buffer were placed in a 96-well microplate. The fluorescence kinetic measurements were recorded at λexcitation = 485 nm and λemission = 530 nm at intervals of 5 min for a runtime of 60 min at 37 °C. Ascorbic acid was used as the reference standard. Each measurement was carried out in triplicate and the capacity of the extract for scavenging peroxyl radicals generated by spontaneous decomposition of AAPH was calculated through the inhibition percentage of the difference between the Area Under the Curve (AUC) of fluorescence decay in the presence (*AUC_sample_*) or absence (*AUC_control_*) of the sample by Equation (4).
(4)$$\%Inh=\frac{AUC_{control}-AUC_{sample}}{AUC_{control}}\times 100$$

AUC was calculated by mean Equation (5):(5)AUC=0.5+∑ fif0
where *f*_0_ and *f_i_* are fluorescence measurements at t = 0 min and every 5 min, respectively.

Reactive Nitrogen Species (RNS) scavenging capacity

RNS was evaluated by the nitric oxide (NO^•^) radical scavenging assay of OLE, according to Ho et al. (2010) [53], based on the reaction of Griess. The 96-well microplate filling distribution was as follows: 100 μL of extract sample at different concentrations (250–2500 μg/mL) in EtOH/H_2_O (1:3, *v*/*v*), and 50 μL of SNP (5 mM) in buffer (30 mM pH = 7.5). The mixture was incubated at room temperature for 2 h under direct light. Nitrile ion concentration was measured at 734 nm after the addition of 100 μL of Ascorbic acid was used as the reference standard. Each measurement was carried out in triplicate and the capacity of the extract for scavenging NO^•^ radicals was calculated through inhibition percentage as described in Equation (3).

#### 2.6.3. Anti-Inflammatory Activity Assay

Anti-inflammatory activity was estimated determining Lipoxidase (LOX) inhibitory capacity using a fluorescence-based assay to monitor enzyme kinetics employing fluorescein as a probe according to Whent et al. (2010) [54]. KM was measured to set the substrate. Mixtures of 100 μL of LA as substrate in EtOH/H_2_O (0.25:1, *v*/*v*); 100 μL of extracts at seven different concentrations for each sample (from 21.5 to 215 μg/mL) in EtOH/H_2_O (0.25:1, *v*/*v*); 75 μL of fluorescein (2 μM) in buffer; and 75 μL of LOX enzyme 0.0208 U/μL in buffer (Tris HCl, pH 9.0, 150 mM) were placed in each well. Quercetin and rosemary extract were used as positive reference standards, and a well without the extract sample was used as negative control. The fluorescence kinetic measurement was carried out in a microplate reader at λexcitation = 485 nm and λemission = 530 nm, runtime 15 min at intervals of 1 min and 25 °C set temperature, to reach Vmean of enzymatic reaction. Equation (2) represents the percentage of inhibition of the sample compared with the negative control, where V_1_ and V_0_ are the mean velocity obtained for LOX in the presence and absence of inhibitors. Each measurement was carried out in triplicate and results are expressed as mean ± standard deviation. IC_50_ value represents concentration (µg/mL) of quercetin or olive leaves extracts that produced 50 percent of enzyme inhibition capacity compared with the control (without inhibitors).

#### 2.6.4. Parallel Artificial Membrane Permeability Assay for the Blood–Brain Barrier (PAMPA-BBB)

Parallel artificial membrane permeation assay (PAMPA-BBB) was performed as a prediction in vitro mechanism of BBB permeability according to the method proposed by Di et al. (2003) [55]. Previously, the BBB solution was prepared by dissolving 8 mg of PBL and 4 mg of cholesterol in 600 μL n-dodecane. Ethanolic olive leaves extract was dissolved in buffer (5 mM pH 7.4) as an initial solution at 10 mg/mL (EtOH/buffer, 1:1, *v*/*v*). Then, the filter membrane of the donor microplate was impregnated with 5 μL of BBB solution and the acceptor microplate was filled with 350 μL of buffer. Then, 200 μL of initial solution was added to the donor microplate which was assembled like a sandwich over the acceptor microplate. The microplate was incubated in the dark for 4 h at 37 °C and continuous agitation. After the incubation process, 200 μL were taken from each microplate, placed into a vial, dried, and reconstituted in 50 μL of EtOH to obtain donor and acceptor solutions. Both solutions were injected in the GC-qTOF-MS system to identify and compare the compounds present in both microplates. Permeability across the artificial BBB was calculated according to Chen et al. (2008) [56] in Equation (6):(6)Pe=−ln [1−CA(t)Ce]A∗(1VD+1VA)∗t
where *P_e_* is permeability (cm/s), *V_D_* and *V_A_* are donor and acceptor well volume, and correspond to 0.35 and 0.2 mL, respectively. *C_D_*_(*t*)_ is compound concentration in the donor well at time t and *C_A_*_(*t*)_ is compound concentration in the acceptor well at time t. A is effective filter area = ƒ ∗ 0.3 cm^2^, *t* = incubation time = 14,400 s, and *C_e_* is the concentration in equilibrium calculated by *C_e_* = [*C_D_*_(*t*)_ ∗ *V_D_* + *C_A_*_(*t*)_ ∗ *V_A_*]/(*V_D_* + *V_A_*).

#### 2.6.5. Cell Culture Conditions and Toxicity Assay

The in vitro toxicity evaluation of the SS adsorbate was tested on two different cell lines: human proximal tubular epithelial cells (HK-2) and human THP-1 monocytes (both cell lines from ATCC^®^, Rockville, MD, USA). HK-2 cells were cultured in Dulbecco’s Modified Eagle Medium Nutrient Mixture (DMEM/Ham’s F12 Thermo Fisher, Grand Island, NY, USA) growth medium supplemented with 10% fetal bovine serum (FBS), 2 mM L-glutamine, 100 U/mL penicillin, 100 μg/mL streptomycin, and 1% insulin-transferrin-selenium (ITS) (all from Thermo Fisher) at 37 °C in 95% humidified air containing 5% CO_2_. Cells were plated in 96-well plates at a density of 5 × 10^3^ cells/well and incubated for 24 h. After cell attachment, different concentrations (2.5 to 40 µg/mL) were dissolved in DMSO, added to the cells and then incubated for 24 h. The solvent did not exceed the concentration of 0.4% (*v*/*v*). The viability of the cells was then determined by 3-(4,5-dimethylthiazol-2-yl)-2,5-diphenyltetrazolium bromide (MTT, Sigma-Aldrich, St. Louis, MO, USA) assay [57]. Briefly, the cell culture medium was removed and cells were incubated with 0.5 mg/mL MTT for 3 h at 37 °C. Finally, DMSO was added to solubilize formazan crystals and the absorbance was measured at 570 nm in a plate reader. Relative cell viability was calculated as the absorbance ratio between adsorbates-treated and DMSO-vehicle (control) cells. THP-1 monocytes were grown and maintained as described by Villalva et al. [58]. Briefly, cells were grown in Roswell Park Memorial Institute (RPMI 1640, Thermo Fisher) culture medium supplemented with 10% FBS, 100 U/mL penicillin, 100 µg/mL streptomycin, 2 mM L-glutamine, and 0.05 mM β-mercaptoethanol (Sigma Aldrich) at 37 °C in 95% humidified air containing 5% CO_2_. Then, cells were seeded in 24-well plates at a density of 5 × 10^5^ cells/mL and monocytes were differentiated to macrophages by maintaining the cells with 100 ng/mL of phorbol 12-myristate 13-acetate (PMA, Sigma-Aldrich) for 48 h. Finally, extract was added into the wells at different concentrations (20 and 40 µg/mL), and the viability of the cells was evaluated using the MTT assay as described above.

In all cases, the toxicity of the extract is shown as cell viability, which is expressed as the percentage of living cells compared with controls. All the experiments were performed in triplicate.

#### 2.6.6. Statistical Analysis

Experimental results are given as mean ± standard deviation (SD). Experimental data results were analyzed by ANOVA and means were compared by Tukey’s HSD (SPSS statics V15 IBM, New York, NY, USA). Statistical significance level was considered for *p* values < 0.05. Means labeled with different alphabetical letters in the same column of the table are considered statistically different at 95% confidence level. Principal component analysis (PCA) was carried out using the statistical software The Unscrambler V9.7 (CAMO Software AS, Oslo, Norway). Multivariate data matrix was analyzed after data autoscaling.

## 3. Results and Discussion

### 3.1. Adsorbent-Assisted scCO_2_ Fractionation of Olive Leaves and GC-QTOF-MS Analysis

In order to obtain extracts from olive leaves enriched in different types of terpenes, a dynamic supercritical fluid extraction–adsorption–desorption process was developed as described in [49] using different adsorbents based on silica gel (S60, S60P, S150 and S150P) and adsorbents based on aluminum (AO and ZeAmG). Additionally, in the current work, we have included the study of SS as adsorbent.

SS is usually employed as “inert” material mixed with the sample to avoid preferential paths during compressed fluids extraction processes. However, can SS be considered an inert material for mixing with the samples in SFE? Is there any retention selectivity in this material for the target compounds? To answer these questions, Figure 2 shows the behavior of SS as an adsorbent in terms of total extraction yield and total recovery of terpenes vs. the amount of CO_2_ per gram of olive leaves (S/F ratio).

Despite being considered an inert material highly used in SFE, SS is based on SiO_2_ and contains Si-OH groups, which have some affinity for polar compounds and residual water from the sample. As can be observed in Figure 2, the kinetic trend of the whole process (extraction–adsorption–desorption) shows a sigmoidal curve similar to what was previously described for other samples [59,60]. As proposed by the same authors [59], the initial delay (lagged portion of the curve in Figure 2) that results in an S-shape kinetic curve, is strongly related to an adsorption/desorption process followed by a constant extraction rate which eventually declines as the solute is depleted. In our case, by observing the total recovery of terpenes at 20 min (S/F 6.5), it is clear that most terpenes are adsorbed at the beginning of the process, being desorbed continuously and extracted at a constant extraction rate until 80 min. From 80 to 120 min, the extraction yield increases slightly but is not related to the content of the terpenes since the terpenes recovery drops strongly after 100 min of processing time. This can be due to the simultaneous co-extraction of other components present in the olive leaves while the adsorption of polar and high molecular weight terpenes (mainly C30 terpenes) takes place in the last steps. A global yield of 0.61% is obtained, meaning that 91% of the total extractable material (0.70% under control conditions, for more details see ref [49]) is recovered from the column. In terms of terpenes recovery, only 55% of the total terpenes were eluted during the 120 min dynamic extraction–adsorption–desorption process, while an additional 8% was washed out from the SS with ethanol. It is important to highlight that the composition of the extract recovered after washing the SS with ethanol corresponds to around 98% of C30 terpenes.

Using a similar approach, a total number of 32 natural extracts from olive leaves enriched in terpenoids were studied. This number comes from the 7 adsorbents and control (i.e., with no adsorbent) at 3 points of kinetic extraction tested (*t* = [0–20], [40–60], and [100–120] min), plus, the 7 adsorbates fraction that remained in the adsorbent and were recovered by washing with ethanol as described in the Materials and Methods section, and a global control (total extraction time = 120 min) obtained by SFE without using any adsorbent.

Table 1 summarizes a total of 40 terpenes and terpenoids derivatives identified in this study by GC-QTOF-MS, according to methodology described in a previous work [49]. Compounds were classified into families according to the number of isoprene units involved in the chemical structure, mainly monoterpenoids (C10), sesquiterpenoids (C15), diterpenoids (C20), and triterpenoids (C30).

The 32 extracts obtained in this study were analyzed by GC-QTOF-MS and the results are shown in Table 2 as percentage of contribution of the most representative terpenes (abundance) to the total abundance of terpenes in the different extracts (considering control, adsorbents, and adsorbates). For a better understanding on the effect of adsorbents on the extraction selectivity, Table 2 is color-coded: shaded in red, yellow, and green to represent a low, medium, and high percentage of contribution, respectively. As it can be seen, two triterpenes (erythrodiol and uvaol) represent more that 70% of abundance in adsorbates compared with other fractions. In addition, the abundance of these triterpenes in the adsorbates increased compared with CL (global control), from around 30% for uvaol in CL to 50% in SS adsorbate (SS ads). The enrichment of this kind of pentacyclic triterpenes is significant considering their potential anti-inflammatory, antioxidant, and cardioprotective bioactivities, among others [12,61,62,63]. α-tocopherol is another compound of interest not only for its recognized antioxidant capacity but also for its ability to transport target compounds with bioactive potential through neuronal cells [64]. According to Table 2, the contribution of α-tocopherol in all adsorbates is similar to the one observed in CL. This is probably due to the low affinity between the compound (lipophilic nature) and the solvent employed to clean the adsorbent (ethanol). Nevertheless, the enrichment of α-tocopherol (C20) along time fractionation is remarkable compared with the control; in this sense, from 1.74% contribution in CL, time fractionation allowed reaching levels of 7.75%, 3.15%, and 2.57% after 20 min (C-20), 60 min (C-60), and 120 min (C-120), respectively. This is even more significant in fractions at 60 min using adsorbents such as S150, S150P, S60, S60P, and AO compared with control (C-60) that showed higher α-tocopherol recovery percentages (15.62, 19.24, 30.37, 17.33, and 29.48%). On the other hand, SS and ZeAmG had a low contribution of α-tocopherol (4.82 and 6.29%) in the same time fraction (60 min). This fact confirms previous results [49] obtained with amorphous and porous silicates (S60, S60P, S150, and S150P) and alumina (AO) that showed an increased recovery of C20, whereas crystalline zeolites and SS favors C30 terpenes recoveries. As for squalene, another important bioactive compound in olive leaves, it is mainly recovered in fractions collected at 120 min for all the silicas studied, representing an increase between 2 to 4-fold compared with the global control; however, an increase of 4.5-fold of squalene in AO is also noticeable. This fact shows the greater affinity of these adsorbents for C10 and C15 terpenes and, therefore, a lower selectivity towards triterpenes.

### 3.2. In Vitro Neuroprotective Potential Assessment of Olive Leaves Extracts

As mentioned, the multifactorial pathophysiology of AD has oriented research towards multi-target strategies to delay the onset and progression of this disease [65]. Following this idea, a multi-target in vitro activity test of the different olive leaves extracts and adsorbates was carried out to evaluate their in vitro potential against AChE, BChE, and LOX enzymes, as well as their antioxidant and radical scavenging properties. Furthermore, the capacity of terpenoid-rich extracts to cross the blood–brain barrier (BBB) was also evaluated. The results of these in vitro bioactivity assays are summarized in Table 3 and Table 4 and are discussed below.

The inhibitory efficiency against AChE of all the extracts obtained in this work is provided in Table 3 as IC_50_ values. As it can be seen, control fractions as well as all the extracts belong to low potency AChE inhibitors capacity group with concentration values above 300 µg/mL, according to the general classification of natural extracts efficacy [22]. Interestingly, it is noticeable that the IC_50_ values against AChE obtained for adsorbates from SS (144.43 ± 29.11), S150P (270.66 ± 16.90), and AO (271.54 ± 13.39) belong to moderate potency inhibitors. The activity of the SS adsorbate showed significant difference with control samples and galantamine (*p* < 0.05) but no significant difference with rosemary (natural standard recognized as potent AChE inhibitor [66,67]). According to Section 3.1, 87.73% of total terpenes of this fraction is composed by erythrodiol and uvaol, both involved in neuroprotective activity [12,68,69].

Regarding the ABTS results, Table 3 shows that in general all olive leaves extracts exhibited lower antioxidant capacity compared with Trolox standard. However, it can be also observed that time fractionation exerts an important influence in the antioxidant properties of olive leaves extracts. The best results were obtained for the adsorbates, with antioxidant capacity values IC_50_ of 23.65 (S150) and 32.68 (S150P) and no significant differences (*p* < 0.05) compared with ascorbic acid and rosemary standards. Furthermore, no significant differences were observed between SS adsorbate and C-120 fractions, characterized by a high content of triterpenes.

The anti-inflammatory potential of the 32 extracts was determined by means of the LOX enzyme inhibition assay and the results are provided in Table 3. The results of the LOX assay of all olive leaves extracts showed low LOX inhibitory capacity compared with quercetin, used as reference. Interestingly, anti-inflammatory results exhibited similar behavior than antioxidant results, with the adsorbates again showing lower IC_50_ values (from 84.29 ± 5.82 to 139.82 ± 11.75).

Additionally, reactive oxygen/nitrogen species (ROS and RNS) were tested for SS adsorbate extract, since this is the most representative extract in terms of AChE inhibition activity and one of the best extracts in terms of antioxidant and anti-inflammatory activities. As shown in Table 4, the SS extract showed potential ROS scavenging properties with IC_50_ =18.27 ± 0.46, although this value is still far from ascorbic acid used as a control (IC_50_ = 1.29 ± 0.09). In terms of RNS, the studied extract exhibited higher scavenging capacity with (IC_50_ = 1036.86 ± 114.21) providing a similar result compared with ascorbic acid used as standard (IC_50_ = 1100.90 ± 13.96). Moreover, for the SS adsorbate, inhibition of BChE was assessed for the selected extract since the palliative treatment of AD consists of increasing acetylcholine levels through the dual inhibitor of AChE and BchE [70]. The results obtained were in line with those previously achieved in our research group for orange juice by-products [50] and suggest the potential of the SS adsorbate as a neuroprotective extract.

The current method based on adsorbent-assisted SFE as proposed in this work, seems to promote the enrichment in some terpenoids in the adsorbates and the elution of other matrix compounds into the fractions. For instance, extracts are highly enriched in tocopherols that were retained in the adsorbent as was described by other authors [71,72]. On the other hand, pentacyclic triterpenes from olive leaves present in adsorbates such as oleanolic and ursolic acids as well as uvaol, β-amyrin, and α-amyrin play an important role in preventing the oxidation produced by free radicals as previously described [7,73]. Several studies reported antioxidant properties of bioactive triterpenes from olive oil by-products decreasing effects of ROS, proteins nitration, and other effects related to oxidative stress [8,73,74,75,76,77]. The obtained adsorbates can be considered good candidates for evaluation of their multi-target neuroprotective activity and, therefore, SS adsorbate was selected for the next step: to study its blood–brain barrier permeability.

### 3.3. Terpenoids Blood–Brain Barrier Permeability Assay

Based on the above results of potential AChE and BChE inhibitory capacity and promising antioxidant/anti-inflammatory activities, the permeability (log Pe) of SS adsorbate fraction was evaluated in order to determine prediction of passive BBB permeation. Results are shown in Table 5.

As can be seen, diterpenes and triterpenes are the main terpenes family involved in BBB permeability for SS adsorbate. Within C20, the role played by hexahydrofarnesyl acetone and tocopherols group (β and α-Tocopherol), whose lipophilic character promotes their BBB permeation, is noticeable. Our results are in agreement with Sánchez [50] who concluded that tocopherols and phytosterols present in orange by-products can cross the BBB, although less efficiently than hydrocarbons mono- and sesquiterpenoids. On the other hand, Ferri [64] concluded the ability of α-tocopherol to promote the transport of the flavonoids across the BBB, favoring reduction of oxidative stress. The presence and Log Pe value of β-Amyrin is remarkable considering the recognized anti-inflammatory potential of this compound [8,78,79]. Our results showed a promising permeability across the BBB compared with, for instance, pharmacological drugs such as galantamine (log Pe cm/s ± SD: −5.35 ± 0.02) or Quercetin (log Pe cm/s ± SD: −7.02 ± 0.08) [80].

### 3.4. Cell-Based Toxicity Evaluation

Cell-based in vitro assays can provide valuable information about biological activities of specific compounds, and they can even predict their behavior in more complex models such as animals or humans. Among the available models, the HK-2 cell line has been considered as a suitable model to predict in vitro toxicity in humans [81]; therefore, this model was selected in our study to evaluate the toxicity effect of SS adsorbate as a previous and necessary step towards the future development of this extract as a natural neuroprotective agent. As show in Figure 3, the SS fraction showed no toxicity at any concentration tested (up to 40 µg/mL).

Based on these results, a second toxicity assay was performed using THP-1 cells, a cell line widely used to evaluate inflammatory and immune responses [82]. In these experiments, the two highest concentrations of SS adsorbate which showed no toxicity on HK-2 cells (20 and 40 µg/mL) were tested. As shown in Figure 3, all the concentrations tested maintained cell viability at maximum.

The non-toxic concentrations used in our study are relatively high compared with other matrices containing terpenoids. For instance, an extract enriched in meroterpenoids from the brown alga *Cystoseira usneoides* presented toxicity in THP-1 cells when using concentrations above 8 µg/mL [83]. Furthermore, other matrices such as *Colebrookea oppositifolia* (Smith) leaves [84] or *Thymbra capitata* and Thymus species [85] presented a toxicity in the same cells when using concentrations above 2 and 8 µg/mL, respectively. Our results demonstrate that the selected adsorbate obtained in this study does not present in vitro toxicity at relatively high concentrations, and future experiments will be performed to corroborate its neuroprotective activity in other cell and in vivo models.

### 3.5. Linking In Vitro Bioactivity and Terpenoids Composition

In order to better understand the relationship between the bioactivity of the 32 studied extracts and their composition, a multivariate data analysis was carried out, seeking differentially enriched terpenoid compounds in the most active samples. Hence, terpenoid compounds annotated in the profiling analysis (Table 1) were considered as composition variables, whereas antioxidant potential (ABTS^•+^) as well as LOX and AChE inhibitory capacity were deemed as in vitro activity variables. These variables were jointly evaluated in a principal component analysis (PCA). The proposed unsupervised multivariate analysis tool allows the evaluation of the compositional variability of the data to obtain the correlation between variables, if any.

According to the PCA results, graphically displayed in Figure 4, the first three principal components (PC1, PC2, and PC3) explained 73% of the total variance. Thus, AO and S150P extracts are grouped in the first quadrant and positively correlated with mono and sesquiterpenoids such as cymenol, thymol, camphene, eugenol, methyleugenol, nerolidol, and caryophyllene. Extracts obtained with lower particle size silica gel exhibited a negative weight in PC1 and positive weight in PC2 (second quadrant), showing negative correlation with most of the terpenoids, which explains the lower enrichment levels in terpenoids for these extracts, as shown in Table 2.

On the other hand, adsorbate samples showed a negative weight in PC1 or PC2, according to their capacity to retain some selected pentacyclic triterpenoids (e.g., uvaol and ursolic acid derivatives) and some particular apocarotenoids (e.g., 4-oxo-β-isodamascol, 3-hydroxy-β-damascone, cyclohexenone derivative, and isololiolide) that are distributed in the third-fourth quadrant of the PC1-PC2 bi-plot (Figure 4A). Regarding the control samples, they are mainly distributed in the fourth quadrant, showing positive correlation with tocopherols and phytol derivatives. The results obtained from the observed correlations in PCA are in good agreement with the differential terpenoids-enrichment degree in extracts and adsorbents analyzed in Table 2 as discussed above.

According to the correlation structure depicted in the bi-plot in Figure 4B, simultaneous evaluation of both groups of variables—composition and the tested neuroprotective activities (AChE, LOX, and ABTS^•+^)—showed positive correlations between in vitro bioactivities and terpenoid compounds with negative contributions in PC2. Thus, AChE inhibitory activity exhibits negative contribution to PC1 and PC2, showing a correlation with the presence of terpenoids like ursolic acid, uvaol, as well as with chromene derivative and isoliolide. These terpenoids were found at higher levels in adsorbate-like samples with negative values in PC1 (SS ads, S150P ads, AO ads) and PC2 (S60 and S60P), as can be seen in the Bi-plots in Figure 4A,B.

On the other hand, ABTS^•+^ and LOX in vitro bioactivities exhibit negative contribution to PC2 and positive weight in PC1, showing correlation with apocarotenoids such as 4-oxo-β-isodamascol, 3-hydroxy-β-damascone, and cyclohexenone derivatives well as with phytosterols (i.e., campesterol and sitosterol).

Although the hypothesis drawn from the above discussions are based on a statistical approach, the results from the proposed multivariate study suggests the existence of latent variables that allow explanation of the in vitro neuroprotective potential of olive leaves extracts on the basis of their terpenoids enrichment. Indeed, β-amyrin, α-amyrin, and uvaol have been reported as the main compounds in olive leaves [7,8,40] with several associated biological properties such as anti-inflammatory, antimicrobial, antifungal, antiviral, anticancer, and anti-ulcer [40,78,86,87,88]. Potential therapeutic effects of diterpenes were reported by [34,42,89], especially related to the antioxidant activity of tocopherols. Neuroprotective multifunctional properties of triterpenoids (C30) have been reported [68,69,90,91,92,93,94,95], and anti-inflammatory, apoptotic, and antioxidant properties, among others, have been attributed to the presence of oleanolic, ursolic, and maslinic acids; erythrodiol, uvaol, and amyrins.

## 4. Conclusions

A selective fractionation process based on dynamic online coupling of SFE and adsorption/desorption was applied for obtaining fractions enriched in terpenoids from olive leaves (*Olea europaea* L.). This fractionation process provides extracts with different terpenoids compositions and different neuroprotective activity. The compositions of the 32 fractions obtained in this work was determined by GC-QTOF-MS. The correlation of bioactivities and chemical composition shows an important contribution of C30 terpenes (ursolic acid derivatives, erythrodiol and uvaol, among others) on the neuroprotective effect of the extracts, while diterpenoids seem to correlate with the antioxidant and anti-inflammatory activities, thus providing a multitargeted approach to AD. The extract that provides the best neuroprotective results was obtained by eluting the sea sand adsorbate (SS ads) with ethanol; toxicity of the extract was evaluated, showing no toxic potential at high concentrations. Moreover, several terpenoids present in the extract showed a promising permeability across the BBB compared with pharmacological drugs such as galantamine. In summary, the results show the interesting potential of this natural extract that will be corroborated in further neuroprotective studies in vitro and in vivo. Moreover, this study provides additional support and information on the biological neuroprotective potential of secondary metabolites such as terpenes and terpenoids from olive leaves.

## Figures and Tables

**Figure 1 foods-10-01507-f001:**
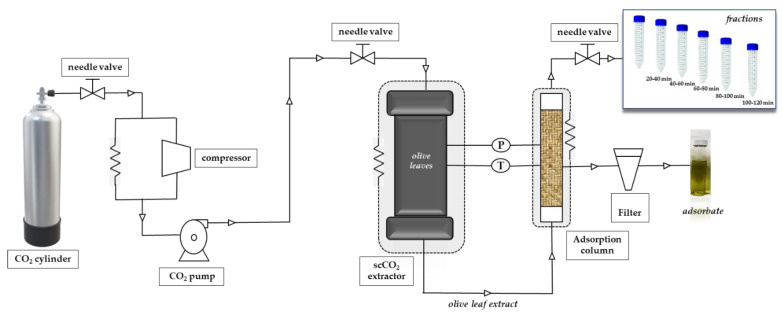
Scheme of the adsorbent-assisted supercritical CO_2_ extraction system.

**Figure 2 foods-10-01507-f002:**
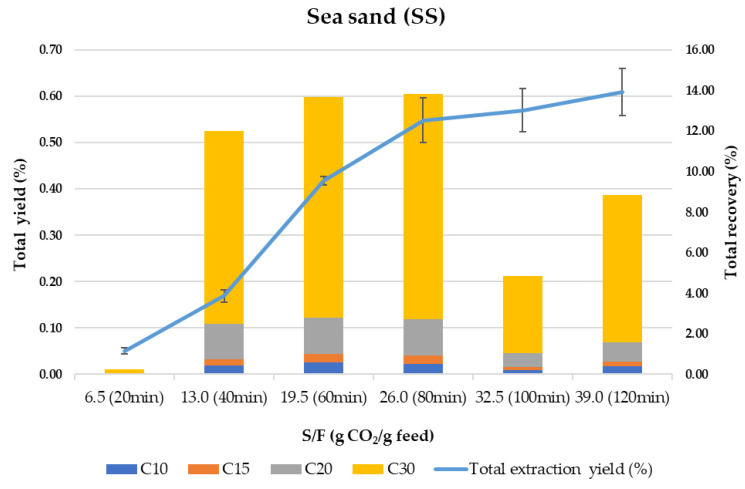
Total extraction yield (%) and total terpenes recovery (%) using sea sand (SS) as adsorbent.

**Figure 3 foods-10-01507-f003:**
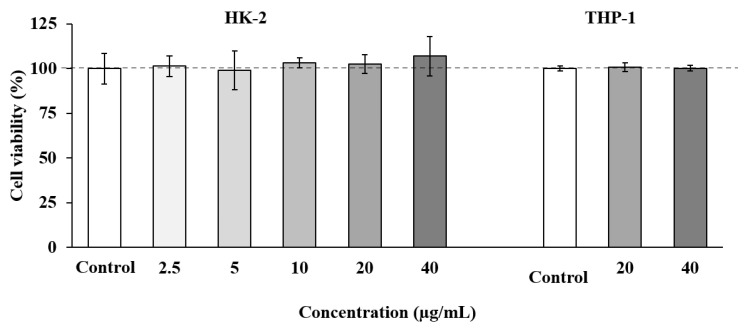
Tubular epithelial cells (HK-2) and human THP-1 monocytes cell viability upon treatment for 24 h with different concentrations of SS adsorbate from olive leaves. Error bars are given as the standard deviation of three independent experiments.

**Figure 4 foods-10-01507-f004:**
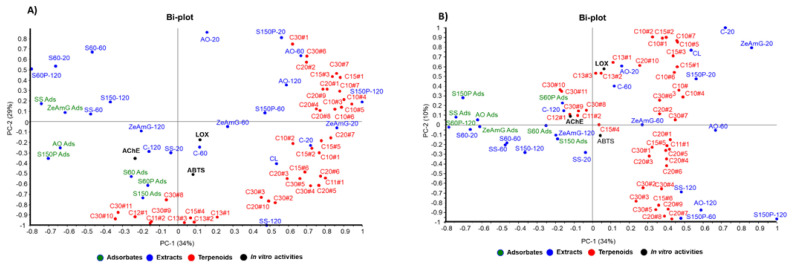
Principal component analysis (PCA) of the 32 extracts from olive leaves, combined with in vitro activities and types of terpenoids (C10, C15, C20, and C30). (**A**) PC-1 and PC-2 biplot, (**B**) PC-2 and PC-3 biplot.

**Table 1 foods-10-01507-t001:** Terpenes and terpenoids in olive leaves extracts identified by gas chromatography-quadrupole-time of flight mass spectrometry (GC-QTOF-MS).

ID	Peak Number	Compound	RetentionTime (Min)	Family
C10#1	1	Borneol isomer	8.35	Monoterpenoid
C10#2	2	Anethole	8.67	Monoterpenoid
C10#3	3	Cymenol isomer	9.90	Monoterpenoid
C10#4	4	Thymol	10.01	Monoterpenoid
C10#5	5	Camphene	10.67	Monoterpene
C10#6	6	Eugenol	10.78	Monoterpenoid
C10#7	7	Methyleugenol	11.37	Monoterpenoid
C15#1	8	Nerolidol	11.95	Sesquiterpenoid
C15#2	9	Farnesene	12.59	Sesquiterpene
C11#1	10	Dihydroactinidiolide	13.14	Apocarotenoid
C15#3	11	Caryophyllene oxide	13.33	Sesquiterpenoid
C13#1	12	4-Oxo-β-isodamascol	13.71	Apocarotenoid
C13#2	13	3-Hydroxy-β-damascone	13.99	Apocarotenoid
C13#3	14	Cyclohexenone derivative	14.38	Apocarotenoid
C15#4	15	γ-Elemene	14.53	Sesquiterpene
C15#5	16	(-)-Globulol	14.62	Sesquiterpenoid
C11#2	17	Isololiolide	15.86	Apocarotenoid
C12#1	18	Chromene derivative	15.99	Meroterpenoid
C20#1	19	Hexahydrofarnesyl acetone	16.43	Diterpenoid
C20#2	20	Geranylgeraniol	17.17	Diterpenoid
C20#3	21	Isophytol	17.49	Diterpenoid
C20#4	22	Phytol	19.07	Diterpenoid
C30#1	23	Squalene	24.73	Triterpene
C20#5	24	Tocospiro A	24.98	Meroditerpenoid
C20#6	25	Tocospiro B	25.14	Meroditerpenoid
C20#7	26	γ-Tocopherol	26.30	Meroditerpenoid
C20#8	27	β-Tocopherol	26.41	Meroditerpenoid
C20#9	28	α-Tocopherol	26.92	Meroditerpenoid
C20#10	29	α-Tocopherolquinone	26.96	Meroditerpenoid
C30#2	30	Stigmasterol	27.69	Triterpenoid
C30#3	31	β-Sitosterol	28.28	Triterpenoid
C15#6	32	Germacrene D	28.59	Sesquiterpene
C30#4	33	β-Amyrin	28.68	Triterpenoid
C30#5	34	α-Amyrin	29.06	Triterpenoid
C30#6	35	β-Amyrin acetate	29.28	Triterpenoid
C30#7	36	Lupenol acetate	29.67	Triterpenoid
C30#8	37	Ursolic acid derivative I	30.60	Triterpenoid
C30#9	38	Ursolic acid derivative II	31.19	Triterpenoid
C30#10	39	Erythrodiol	31.50	Triterpenoid
C30#11	40	Uvaol	32.08	Triterpenoid

**Table 2 foods-10-01507-t002:** Total terpenes contribution (%) of the main terpenes identified in this study in the 32 olive leaves extracts. Color code: Red (low concentration); Green (high concentration).

Compound	Fam	Total Terpenes Contribution (%) *							
		GC	C-20	C-60	C-120	SS-20	SS-60	SS-120	SS Ads
Methyleugenol	C10	0.02	0.04	0.02	0.01	0.00	0.00	0.00	0.00
Anethole	C10	0.06	0.03	0.01	0.01	0.00	0.00	0.00	0.00
Farnesene	C15	0.01	0.01	0.00	0.00	0.00	0.00	0.00	0.00
Isophytol	C20	0.02	0.02	0.02	0.02	0.01	0.01	0.02	0.00
Stigmasterol	C30	0.02	0.02	0.02	0.02	0.02	0.02	0.03	0.00
Hexahydrofarnesyl acetone	C15	0.24	0.33	0.22	0.15	0.50	0.33	0.31	0.17
Germacrene D	C15	2.08	1.83	1.44	1.33	1.32	1.58	1.67	0.30
Tocospiro B	C20	1.75	1.85	1.47	1.14	2.86	2.18	1.48	0.18
α-Amyrin	C30	3.42	2.81	2.42	2.52	2.62	2.09	3.39	0.56
β-Sitosterol	C30	1.31	1.18	1.10	1.05	1.11	1.53	1.58	0.34
α-Tocopherol	C20	1.74	7.75	3.15	2.57	6.03	4.82	7.51	1.49
Squalene	C30	4.42	5.92	2.48	1.37	4.74	3.04	4.64	0.58
β-Amyrin	C30	11.54	11.91	9.34	8.75	10.13	8.80	10.05	1.37
Erythrodiol	C30	29.76	23.93	29.92	31.53	28.59	32.06	27.84	37.43
Uvaol	C30	31.53	30.48	39.08	41.63	32.09	34.85	31.82	50.30
**Compound**	**Fam**	**Total Terpenes Contribution (%) ***							
		**S150-** **20**	**S150-** **60**	**S150-** **120**	**S150** **Ads**	**S150P-** **20**	**S150P-** **60**	**S150P-** **120**	**S150P** **Ads**
Methyleugenol	C10	0.00	0.00	0.00	0.00	0.04	0.02	0.02	0.00
Anethole	C10	0.00	0.00	0.00	0.00	0.02	0.02	0.01	0.00
Farnesene	C15	0.00	0.00	0.00	0.00	0.00	0.01	0.00	0.00
Isophytol	C20	0.00	0.00	0.00	0.01	0.00	0.07	0.03	0.00
Stigmasterol	C30	0.01	0.01	0.01	0.02	0.01	0.04	0.04	0.01
Hexahydrofarnesyl acetone	C15	0.07	0.28	0.33	0.16	0.59	0.33	0.52	0.08
Germacrene D	C15	1.61	1.63	1.17	2.06	1.13	6.04	4.58	0.43
Tocospiro B	C20	0.88	1.43	0.89	1.33	0.61	2.40	1.94	0.46
α-Amyrin	C30	5.01	2.28	1.78	3.24	1.73	9.37	7.50	0.79
β-Sitosterol	C30	0.76	1.07	1.03	1.33	0.47	2.74	2.93	0.88
α-Tocopherol	C20	3.78	15.62	7.77	1.97	21.81	19.24	17.41	1.12
Squalene	C30	6.98	28.05	8.53	0.27	22.72	13.63	16.19	0.24
β-Amyrin	C30	20.18	10.15	7.64	11.23	6.70	26.53	20.04	2.40
Erythrodiol	C30	26.79	15.78	27.99	30.41	12.57	3.62	6.63	37.44
Uvaol	C30	25.54	16.87	36.47	39.66	20.08	6.05	10.92	48.88
**Compound**	**Fam**	**Total Terpenes Contribution (%) ***							
		**S60-** **20**	**S60-** **60**	**S60-** **120**	**S60** **Ads**	**S60P-** **20**	**S60P-** **60**	**S60P-** **120**	**S60P** **Ads**
Methyleugenol	C10	0.00	0.00	0.00	0.00	0.00	0.00	0.00	0.01
Anethole	C10	0.00	0.01	0.00	0.00	0.00	0.00	0.01	0.00
Farnesene	C15	0.00	0.00	0.00	0.01	0.00	0.00	0.01	0.00
Isophytol	C20	0.01	0.00	0.03	0.02	0.00	0.00	0.00	0.01
Stigmasterol	C30	0.01	0.02	0.01	0.02	0.00	0.00	0.02	0.02
Hexahydrofarnesyl acetone	C15	0.51	0.90	0.38	0.08	0.05	0.07	0.25	0.16
Germacrene D	C15	0.58	0.17	0.40	1.62	0.04	0.01	2.16	1.53
Tocospiro B	C20	0.44	0.47	0.75	1.13	0.07	0.10	0.50	1.18
α-Amyrin	C30	1.12	0.44	2.48	2.83	0.14	0.19	3.78	2.45
β-Sitosterol	C30	1.14	1.89	0.70	1.22	0.46	1.08	1.70	1.21
α-Tocopherol	C20	1.07	30.37	14.01	1.53	1.78	17.33	7.24	1.77
Squalene	C30	44.44	39.12	10.78	0.34	1.46	0.92	9.70	0.36
β-Amyrin	C30	3.19	2.19	1.68	9.48	0.17	0.06	12.00	9.22
Erythrodiol	C30	15.50	5.66	5.87	31.17	47.93	38.73	21.04	30.62
Uvaol	C30	28.36	8.61	44.69	43.44	42.36	38.08	36.81	42.90
**Compound**	**Fam**	**Total Terpenes Contribution (%) ***							
		**AO-** **20**	**AO-** **60**	**AO-** **120**	**AO** **Ads**	**ZeAmG-** **20**	**ZeAmG-** **60**	**ZeAmG-** **120**	**ZeAmG Ads**
Methyleugenol	C10	0.07	0.04	0.02	0.00	0.04	0.02	0.01	0.00
Anethole	C10	0.01	0.01	0.01	0.00	0.05	0.02	0.01	0.01
Farnesene	C15	0.01	0.01	0.00	0.00	0.01	0.00	0.00	0.00
Isophytol	C20	0.00	0.00	0.00	0.00	0.04	0.03	0.01	0.01
Stigmasterol	C30	0.01	0.01	0.02	0.02	0.02	0.02	0.02	0.01
Hexahydrofarnesyl acetone	C15	1.46	0.85	0.81	0.10	0.64	0.41	0.19	0.17
Germacrene D	C15	0.83	1.18	3.69	1.33	2.42	2.22	1.64	0.87
Tocospiro B	C20	0.53	3.79	3.77	0.61	2.70	1.50	0.71	0.24
α-Amyrin	C30	1.31	1.40	6.00	2.04	3.46	3.25	2.88	1.07
β-Sitosterol	C30	1.12	0.73	1.39	1.63	1.28	1.33	1.23	1.13
α-Tocopherol	C20	3.40	29.48	24.50	2.06	7.72	6.29	4.79	2.27
Squalene	C30	37.84	28.75	18.42	0.14	13.61	8.23	4.07	0.76
β-Amyrin	C30	4.71	6.65	21.99	8.17	15.99	13.16	9.70	3.73
Erythrodiol	C30	15.03	5.26	2.21	50.32	17.90	25.24	30.40	36.96
Uvaol	C30	21.44	7.55	3.30	24.95	17.66	27.21	37.25	47.63

* Total terpenes contribution = (terpenoid abundance/total abundance) ∗ 100; CL: global control; C-20, C-60, C-120: controls at 20, 60, and 120 min; SS-20, SS-60, SS-120, SS ads: sea sand at 20, 60, 120 min and adsorbate;; S150-20, S150-60, S150-120, S150 ads: silica 150Å, mesh (250–500 µm) at 20, 60, 120 min and adsorbate. S150P-20, S150P-60, S150P-120, S150P ads: silica 150Å, mesh (35–70 µm) at 20, 60, 120 min and adsorbate; S60-20, S60-60, S60-120, S60 ads: silica 60Å, mesh (250–500µm) at 20, 60, 120 min and adsorbate. S60P-20, S60P-60, S60P-120, S60P ads: silica 60Å, mesh (40–63 µm) at 20, 60, 120 min and adsorbate; AO-20, AO-60, AO-120, AO ads: aluminum oxide at 20, 60, 120 min and adsorbate.; ZeAmG-20, ZeAmG-60, ZeAmG-120, ZeAmG ads: Zeolite Y, ammonium at 20, 60, 120 min and adsorbate.

**Table 3 foods-10-01507-t003:** IC_50_ values (µg/mL) for AChE, ABTS^•+^, and LOX assays obtained for the 32 olive leaves extracts and reference-standards studied.

No.	SAMPLES	AChE Inhibitory Capacity	Antioxidant Capacity by ABTS	Anti-Inflammatory Capacity by LOX
Olive leaves extracts used as control fractions				
01	Global control	408.41 ± 0.92 ^defg^	108.67 ± 1.88 ^i^	153.79 ± 0.55 ^hij^
02	Control-20 min	440.52 ± 24.43 ^efgh^	91.58 ± 0.44 ^hijk^	107.30 ± 2.44 ^bcdef^
03	Control-60 min	487.37 ± 34.20 ^f^	95.24 ± 1.79 ^hij^	187.75 ± 9.21 ^jk^
04	Control-120 min	405.88 ± 22.30 ^defg^	67.29 ± 0.61 ^efg^	120.64 ± 8.26 ^defg^
Olive leaves extracts obtained with silicas				
05	SS-20 min	436.68 ± 32.29 ^efgh^	62.67 ± 0.81 ^def^	103.30 ± 9.86 ^bcde^
06	SS-60 min	455.97 ± 44.87 ^fg^	177.16 ± 1.02 ^k^	159.58 ± 15.26 ^hi^
07	SS-120 min	300.80 ± 20.52 ^bc^	46.43 ± 1.32 ^bcd^	225.09 ± 9.18 ^k^
08	S150–20 min	322.88 ± 5.90 ^bc^	134.95 ± 0.99 ^jkl^	83.53 ± 7.16 ^ab^
09	S150–60 min	746.99 ± 35.71 ^hi^	258.49 ± 11.49 ^l^	516.51 ± 13.99 ^m^
10	S150–120 min	577.57 ± 25.88 ^g^	326.48 ± 9.09 ^m^	548.87 ± 68.79 ^n^
11	S150P-20 min	482.28 ± 34.32 ^f^	83.07 ± 0.31 ^ghij^	110.80 ± 8.93 ^cdef^
12	S150P-60 min	483.43 ± 28.23 ^f^	101.48 ± 2.16 ^hi^	119.24 ± 6.68 ^defg^
13	S150P-120 min	453.04 ± 36.93 ^fg^	104.72 ± 5.87 ^i^	171.79 ± 29.01 ^ij^
14	S60–20 min	805.47 ± 103.47 ^k^	320.67 ± 4.25 ^n^	n.d.
15	S60–60 min	761.14 ± 49.23 ^hi^	1789.61 ± 27.56 ^q^	201.40 ± 14.28 ^kl^
16	S60–120 min	790.58 ± 37.83 ^h^	59.30 ± 0.42 ^cde^	n.d.
17	S60P-20 min	788.43 ± 46.68 ^h^	501.75 ± 10.06 ^o^	n.d.
18	S60P-60 min	n.d.	391.33 ± 54.50 ^m^	n.d.
19	S60P-120 min	712.53 ± 49.97 ^h^	953.40 ± 23.52 ^p^	171.86 ± 11.01 ^ij^
Olive leaves extracts obtained with aluminum oxide				
20	AO-20 min	549.49 ± 51.36 ^g^	191.74 ± 0.19 ^k^	169.17 ± 10.65 ^ij^
21	AO-60 min	463.82 ± 26.86 ^fg^	39.99 ± 0.03 ^abc^	75.20 ± 9.80 ^a^
22	AO-120 min	449.50 ± 39.71 ^fgh^	79.98 ± 0.40 ^fghi^	319.76 ± 3.12 ^l^
Olive leaves extracts obtained with zeolites				
23	ZeAmG-20 min	418.40 ± 16.96 ^efg^	128.49 ± 0.45 ^j^	83.27 ± 9.23 ^ab^
24	ZeAmG-60 min	391.84 ± 34.09 ^def^	130.38 ± 2.96 ^jk^	192.65 ± 7.00 ^jk^
25	ZeAmG-120 min	417.21 ± 35.89 ^defg^	153.55 ± 0.04 ^j^	100.54 ± 3.98 ^abcd^
Olive leaves extracts in the different adsorbates				
26	SS	144.43 ± 29.11 ^a^	82.59 ± 1.08 ^ghij^	104.82 ± 11.40 ^bcdef^
27	S150	420.78 ± 11.98 ^efg^	23.65 ± 0.11 ^a^	84.29 ± 5.82 ^abc^
28	S150P	270.66 ± 16.90 ^b^	32.68 ± 0.11 ^ab^	139.82 ± 11.75 ^ghi^
29	S60	383.41 ± 28.25 ^de^	144.61 ± 0.33 ^jkl^	96.94 ± 7.20 ^abcd^
30	S60P	357.67 ± 16.71 ^cd^	73.00 ± 1.63 ^efgh^	104.34 ± 6.57 ^bcde^
31	AO	271.54 ± 13.39 ^b^	148.22 ± 3.46 ^jk^	131.88 ± 4.02 ^fgh^
32	ZeAmG	447.64 ± 43.20 ^fgh^	54.26 ± 0.39 ^cde^	129.56 ± 10.29 ^efgh^
Standards				
	Galantamine *	0.40 ± 0.02 ^l^	--	--
	Trolox *	--	2.50 ± 0.02 ^r^	--
	Ascorbic acid *	--	25.00 ± 0.30 ^a^	--
	Quercetin *	--	--	19.71 ± 0.24 ^p^
	Rosemary **	107.85 ± 8.39 ^a^	35.63 ± 1.14 ^abc^	9.82 ± 0.88 ^o^

Results are expressed as the mean ± SD (*n* = 3). * Chemical standard; ** Reference natural extract; n.d.: no data. Different letters in the same column show significant statistical differences (*p* < 0.05).

**Table 4 foods-10-01507-t004:** IC_50_ values (µg/mL) from BChE, ROS, and RNS from sea sand (SS) adsorbate.

Sample	BChE	ROS	RNS
Sea sand adsorbate	183.82 ± 22.47 ^a^	18.27 ± 0.46 ^a^	1036.86 ± 114.21 ^a^
Galantamine *	2.36 ± 0.02 ^b^	--	--
Ascorbic acid *	--	1.29 ± 0.09 ^b^	1100.90 ± 13.96 ^b^
Trolox *	--	0.98 ± 0.11 ^c^	--

Results are expressed as the mean ± SD (*n* = 3). * Chemical standard. Different letters in the same column show significant statistical differences (*p* < 0.05).

**Table 5 foods-10-01507-t005:** Main terpenes from SS adsorbate identified in PAMPA-BBB study.

Terpene	PAMPA-BBB Log Pe (cm/s) (RSD %) Sea Sand Adsorbate
Chromene derivative	−6.82 (8.2)
Hexahydrofarnesyl acetone	−4.39 (12.1)
Phytol	−6.12 (26.6)
Squalene	−4.88 (14.9)
γ-Tocopherol	−5.05 (13.2)
β-Tocopherol	−4.97 (11.8)
α-Tocopherol	−5.94 (10.1)
β-Sitosterol	−6.04 (7.8)
β-Amyrin	−6.32 (3.1)
Lupenol acetate	n.d. *
Ursolic acid derivative I	n.d. *
Ursolic acid derivative II	−5.61 (4.9)
Erythrodiol	−6.24 (18.5)
Uvaol	−5.82 (18.4)

* n.d.: no data.

## Data Availability

Please refer to suggested Data Availability Statements in section “MDPI Research Data Policies” at https://www.mdpi.com/ethics (accessed on 28 June 2021). You might choose to exclude this statement if the study did not report any data.

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
