# Peer review of "Neuroprotective Effect of Terpenoids Recovered from Olive Oil By-Products"

_foods, 2021, doi:10.3390/foods10071507_

Round 1

Reviewer 1 Report

Interesting study using waste from production of olive oil. 

Questions arise about the form and method of application of the by-products (food, medicine ...).
Why did the authors choose Alzheimer's disease? Are there any premises for this?
Can other indications be given?

Author Response

Interesting study using waste from production of olive oil.

We want to thank Reviewer 1 for considering our study interesting.

Questions arise about the form and method of application of the by-products (food, medicine ...).

Although there are many possibilities for the application of the by-products, the work is focused in the development of food ingredients with neuroprotective activity and with strong scientific basis that could be used for the manufacturing of food/dietary supplements or nutraceuticals. In this sense, it is important to consider the global increase of the nutraceutical industry in the past decade which market has grown from a value of USD 140 Billion in 2010 to USD 382 Billion in 2020 and is expected to reach USD 486 Billion by 2025. The rapid expansion of this sector is attributed to the shift in the healthcare habits from treatment to prevention. Therapists and patients are now thinking differently about health and disease and consequently adopt dietary alternatives that can target risk factors and physiological pathways to prevent the onset of chronic diseases. More specifically, nutraceuticals for neurological disorders (Makkar et al., Int. J. Mol. Sci. 21 (2020) 4424) are recognized a huge market (globally, more than 10 million people suffer from neurological disorders annually). As a result, the interest in developing new supplements based on natural extracts that can help ameliorating neurodegeneration is constantly increasing. In this sense, it is worth mentioning the interest of nutraceutical compounds for a “multitarget” therapy of AD, that has been recently recognized as one important possibility for the future control of this disease (Guzmán-Martínez et al., J. Alzheimer Dis., Pre-press (2021) 1-13).

Why did the authors choose Alzheimer's disease? Are there any premises for this?

The selection of Alzheimer Disease as target in this study has different reasons: first of all, considering the social and economic impact of this disease worldwide (and in particular in Spain), we have been working in the last few years in a research project focused on investigating the terpenoids fraction of different food by-products, plants, algae and microalgae, to develop specific green extraction conditions for this type of compounds, to characterize them chemically and to investigate their bioactivity as a multitarget neuroprotective treatment of Alzheimer disease. To do this, we first accept that there is not cure for this pandemic yet and second, that the bioactivity of terpenoid compounds is different depending on their size and nature (e.g., antioxidant, anti-inflammatory, anti-cholinergic, etc), so, a combination of these compounds can provide the desirable multitarget approach, once their pass through the blood brain barrier has been demonstrated.

Why selecting terpenoids as target compounds is related to the diversity of structures found in nature on these types of compounds, on their associated potential neuroprotective effect on Alzheimer’s disease (Akaberi, M.;et al. (2016) Phyther. Res., 30, 878–893; reference 89 in the manuscript), and on the lack of information on the molecular mechanisms behind this activity. In this sense, as a source of terpenoids, olive leaves can provide all types of structures, ranging from C10-C40; some of them (short-chain terpenes (C10-C15) have been described in the volatile composition of olive leaves (Flamini et al.,2003)). Among all terpenes present in olive leaves, triterpenes (C30), carotenoids (C40) and sterols (C30) are the most promising in terms of bioactivities. For instance, the most abundant triterpenoids in olive leaves (such as those derived from oleanane and ursane) have been described as possessing pharmacological activity against neurodegenerative disorders (Ruszkowski, P. et al., (2014) Mini Rev Org Chem, 11(3), 307-315).

Can other indications be given?

Indeed, many other compounds have been described in olive leaves with potential bioactivity against several diseases. As mentioned in the introduction of the manuscript, “The bioactivity of olive leaves has been traditionally associated to its content in phenolic derivatives [4–6] and flavonoids and terpenoids [7–10]. These compounds provide a wide range of healthy properties [11–13], such as anti-inflammatory [12,13], antioxidant [11,14–16], and antiproliferative against cancer cell lines [11,17].”, but, in this particular case, only terpenes were targeted since this was the main hypothesis of our research project.  

Reviewer 2 Report

The paper is written in an appropriate language style. However, the text contains a relevant number of small inconsistencies, it seems it was written in a hurry: some of them are here indicated but the suggestion is to check again the whole text. Some minor revisions are here suggested:

  • Figure 1: please improve the quality of the text.
  • Line 176-177: it is no clear to me if the DB-5 column length is 20 m + 10 of Duragard column for a total of 30 m, or DB-5 is 30 m + 10 of Duragard column.
  • Line 184-187: please indicate the threshold of similarity to the NIST MS database selected to consider a compounds identified. Why linear retention index were not used? Especially using a non-polar phase for the GC column they would be useful to confirm the identification.
  • Line 359: substitute "Figure 1" with "Figure 2".
  • Line 454: change "representaive" in "representative". Please check the whole text for similar inconstistencies.
  • Table 2: please change format number from i.e., "0,02" to "0.02" by using points instead of commas.
  • Line 656: eliminate the double comma.

Author Response

The paper is written in an appropriate language style. However, the text contains a relevant number of small inconsistencies, it seems it was written in a hurry: some of them are here indicated but the suggestion is to check again the whole text.

Authors want to thank reviewer 2 for his/her comments on the manuscript. We have carefully checked for inconsistencies and corrected them. All modifications have been higlighted in red in the last version of the manuscript.   

Some minor revisions are here suggested:

Figure 1: please improve the quality of the text.

We have completely modified description of Figure 1, as follows: “The adsorbent-assisted supercritical CO2 extraction process was carried out at 30 MPa and 60 °C dynamically for 120 min. The adsorbent material was placed in a stainless-steel cylindrical adsorption cell (29 cm length and 0.65 cm i.d., for a total column volume of 38.5 cm3) after the extraction cell, as shown in Figure 1; adsorbents were packed into the adsorption column with glass wool; high-quality cellulose disk filters were located at the entrance and exit of the column to prevent plugging. Carbon dioxide passed through the supercritical extraction cell and the extracted solute(s) was adsorbed dynamically by the packed material in the adsorption column; the whole process was carried out at the same P and T, that is, 30 Mpa and 60 °C. Fractions were collected every 20 min at the exit of the adsorption column, after depressurization through an expansion valve (Parker Autoclave Engineers, Erie, PA, USA). After 120 min, complete depressurization of the system was carried out for a total of 30 min.”

Line 176-177: it is no clear to me if the DB-5 column length is 20 m +10 of Duragard column for a total of 30 m, or DB-5 is 30 m + 10 of Duragard column.

Information has been clarified as follows: “The separation was carried out using an Agilent Zorbax DB5- MS Column (30 m × 250 μm i. D. x 0.25 μm) + 10 m Duragard Capillary Column”

Line 184-187: please indicate the threshold of similarity to the NIST MS database selected to consider a compounds identified. Why linear retention index were not used? Especially using a non-polar phase for the GC column they would be useful to confirm the identification.

The following information has been added to provide more details on the identification process: “Target terpenes were annotated by systematic mass spectra deconvolution and search in MS database, using Agilent Mass Hunter Unknown Analysis tool and NIST MS database Search. A total of 40 terpenes and terpenoids were tentatively identified on the basis of the positive match of the experimental mass spectra with MS databases (i.e., NIST and Fiehn lib), exact mass values as determined by HRMS, data reported in literature, and commer-cial standards when available. GC-QTOF-MS parameters such as retention time, generated molecular formula, match factor values from MS database search and main HRMS fragments were considered for annotation. Identification reliability was considered satis-factory for chemical structures, showing math factor values above 70. Terpenoids such as thymol, squalene, phytol, alpha-tocopherol, alpha-amyrin, uvaol and erytrhrodiol were confimed with reference standard. More information about the structural elucidation of target terpenoids can be found in our recently reported paper by Suárez et al. (2021) [49].”

Line 359: substitute "Figure 1" with "Figure 2".

Done.

Line 454: change "representaive" in "representative". Please check the whole text for similar inconstistencies.

Done. Small inconsistencies have been carefully checked.

Table 2: please change format number from i.e., "0,02" to "0.02" by using points instead of commas.

We sincerely thank reviewer for carefully checking this. We have revised all Figures and Tables and change formats accordingly (using points instead of commas).

Line 656: eliminate the double comma.

Done.